# Collider bias undermines our understanding of COVID-19 disease risk and severity

Gareth J. Griffith [1,2,4], Tim T. Morris [1,2,4], Matthew J. Tudball [1,2,4], Annie Herbert[1,2,4], Giulia Mancano[1,2,4], Lindsey Pike[1,2], Gemma C. Sharp [1,2], Jonathan Sterne[2], Tom M. Palmer [1,2], George Davey Smith [1,2], Kate Tilling [1,2], Luisa Zuccolo[1,2], Neil M. Davies [1,2,3] & Gibran Hemani [1,2,4 ✉]

Numerous observational studies have attempted to identify risk factors for infection with SARS-CoV-2 and COVID-19 disease outcomes. Studies have used datasets sampled from patients admitted to hospital, people tested for active infection, or people who volunteered to participate. Here, we highlight the challenge of interpreting observational evidence from such non-representative samples. Collider bias can induce associations between two or more variables which affect the likelihood of an individual being sampled, distorting associations between these variables in the sample. Analysing UK Biobank data, compared to the wider cohort the participants tested for COVID-19 were highly selected for a range of genetic, behavioural, cardiovascular, demographic, and anthropometric traits. We discuss the mechanisms inducing these problems, and approaches that could help mitigate them. While collider bias should be explored in existing studies, the optimal way to mitigate the problem is to use appropriate sampling strategies at the study design stage.

[1] Medical Research Council Integrative Epidemiology Unit, University of Bristol, Bristol BS8 2BN, UK. [2] Population Health Sciences, Bristol Medical School, University of Bristol, Oakfield House, Oakfield Grove, Bristol BS8 2BN, UK. [3] K.G. Jebsen Center for Genetic Epidemiology, Department of Public Health and Nursing, NTNU, Norwegian University of Science and Technology, Trondheim, Norway. [4] These authors contributed equally: Gareth J. Griffith, Tim T. Morris, Matthew J. Tudball, Annie Herbert, Giulia Mancano, Gibran Hemani. ✉email: g.hemani@bristol.ac.uk

Health care providers, researchers and private companies, amongst others, are generating data on the COVID-19 disease status of millions of people to understand the risk factors relevant to SARS-CoV-2 in the general population (defined in Box 1). Numerous studies have reported risk factors associated with COVID-19 infection and subsequent disease severity, such as age, sex, occupation, smoking and ACE-inhibitor use[1–10]. But to make reliable inference about the causes of infection and disease severity, it is important that the biases which induce spurious associations in observational data are understood and assessed. Bias due to confounding remains well-understood and attempts to address it are typically made (bar rare exceptions e.g. ref. [11]). But the problem of collider bias (sometimes referred to as selection bias, sampling bias, ascertainment bias, Berkson's paradox) has major implications for many published studies of COVID-19 and is seldom addressed.

A collider is most simply defined as a variable that is influenced by two other variables, for example when a risk factor and an outcome both affect the likelihood of being sampled (they "collide" in a Directed Acyclic Graph, Fig. 1a). Colliders become an issue when they are conditioned upon in analysis, as this can distort the association between the two variables influencing the collider. Importantly, it is possible to distort the association between two variables that do not directly influence the collider (Fig. 1b). If the factors that influence sample selection themselves influence the variables of interest, the relationship between these variables of interest can become distorted. This is sometimes referred to as M-bias due to the shape of the Direct Acyclic Graph[12].

Collider bias can arise when researchers restrict analyses on a collider variable[13–15]. Within the context of COVID-19 studies, this may relate to restricting analyses to those people who have experienced an event such as hospitalization with COVID-19, been tested for active infection or who have volunteered their participation in a large scale study (Fig. 2a). Among hospitalized patients, the relationships between any variables that relate to hospitalization will be distorted compared to among the general population. The magnitude of this distortion can be large, inducing associations that do not exist in the general population or attenuating, inflating or reversing the sign of existing associations[16]. As such, associations based on ascertained COVID-19 datasets may not reflect patterns in the population of interest (i.e.

lack of external validity). Furthermore, when attempting to draw causal inferences from ascertained datasets, such effects may not even be valid within the dataset itself (i.e. lack of internal validity) (Box 1). This is because associations induced by collider bias are properties of the sample, rather than the individuals that comprise it, so the associations estimated using the sample will not be a reliable indication of the individual level causal effects. Collider bias, therefore, causes associations to fail to generalise beyond the sample and for causal inferences to be inaccurate even within the sample. It is this second characteristic which distinguishes collider bias within the more general concept of selection bias. Selection bias can occur when there are effect modifiers that are distributed differently in the sample than in the population, thus causing effects to differ between the two. However, while this limits the generalisability of causal effects on the population, those effects are valid within the sample[17].

As illustration, consider the hypothesis that being a health worker is a risk factor for severe COVID-19 disease. Under the assumption of a higher viral load due to their occupational exposure, healthcare workers will on average experience more severe COVID-19 symptoms compared to the general population. The target population within which we wish to test this hypothesis is adults in any occupation (or unemployed); the exposure is being a health worker the outcome is COVID-19 symptom severity. The only way we can reliably estimate COVID-19 status and severity is by considering individuals who have a confirmed positive polymerase chain reaction (PCR) test for COVID-19. However, restrictions on the availability of testing especially in the early stages of the pandemic mean that the available study sample is necessarily restricted to those individuals who have been tested for active COVID-19 infection. If we take the UK as an example (until late April 2020), let us assume a simplified scenario where all tests were performed either on frontline health workers (as critical vectors for disease among high-risk individuals), or members of the general public who had symptoms severe enough to require hospitalisation (as high-risk individuals). In this testing framework, our sample of participants will have been selected for both the hypothesised risk factor (being a healthcare worker) and the outcome of interest (severe symptoms). Our sample will therefore contain all health workers who are tested regardless of their symptom severity, while only non-health workers with severe symptoms will be included. In

---

**Box 1 | Collider bias in the context of aetiological and prediction studies**

The term "risk factor" has been used synonymously for both causal factors and predictors in the literature[79,80]. An aetiological study seeks to identify causes of the outcome of interest ("causal factors"), whereas a predictive study aims to develop scores that predict the outcome from a range of variables ("predictors") which need not be causal. While the term 'risk factor' can be ambiguous and refer to either a hypothesised causal determinant or a predictor of the disease, we use it throughout this paper for the sake of brevity as causal inference and prediction analyses both share a vulnerability to the detrimental impacts of collider bias in the COVID-19 context—where typically the selected samples are being used to develop models relevant to the general population. But for clarity we outline below how collider bias differentially impacts causal inference and prediction.

Risk factors measured in observational studies may associate with outcomes of interest (e.g. hospitalised with COVID-19), for many reasons. For example, the factor may affect the outcome (true causal interpretation), statistical evidence of association may be purely due to chance, the outcome may affect the factor (reverse causation), there may be a third factor that causes both the exposure and the outcome (confounding), or the exposure and outcome (or causes of the exposure and/or outcome) may influence the likelihood of being selected into the study (collider bias).

Aetiological studies are in principle only concerned with the causal effect and aim to avoid all forms of bias. By contrast, some forms of bias such as confounding or reverse causation can actually improve the performance of a prediction study. As long as the causal structure by which the study sample is drawn from the target population is the same as in the population in which predictions will be made, it can be of benefit to leverage these distinct association mechanisms to improve prediction accuracy[81,82].

Similarly, under certain circumstances, collider bias can improve prediction performance if the training sample and the sample to be predicted have the same patterns of sample selection. For example, if the factors causing having a test for COVID-19 are the same/similar across the UK, a predictive model for the result being positive that was developed in London will perform well in the North East of England if those samples are both non-randomly selected in the same way. However, collider bias is a problem for the generalisability of both causal inference and prediction in the target population when the training sample is non-randomly selected, because it induces artefactual associations that are idiosyncratic to that dataset. If the intention is to predict COVID-19 status, rather than COVID-19 status conditional on being tested, the prediction will underperform.

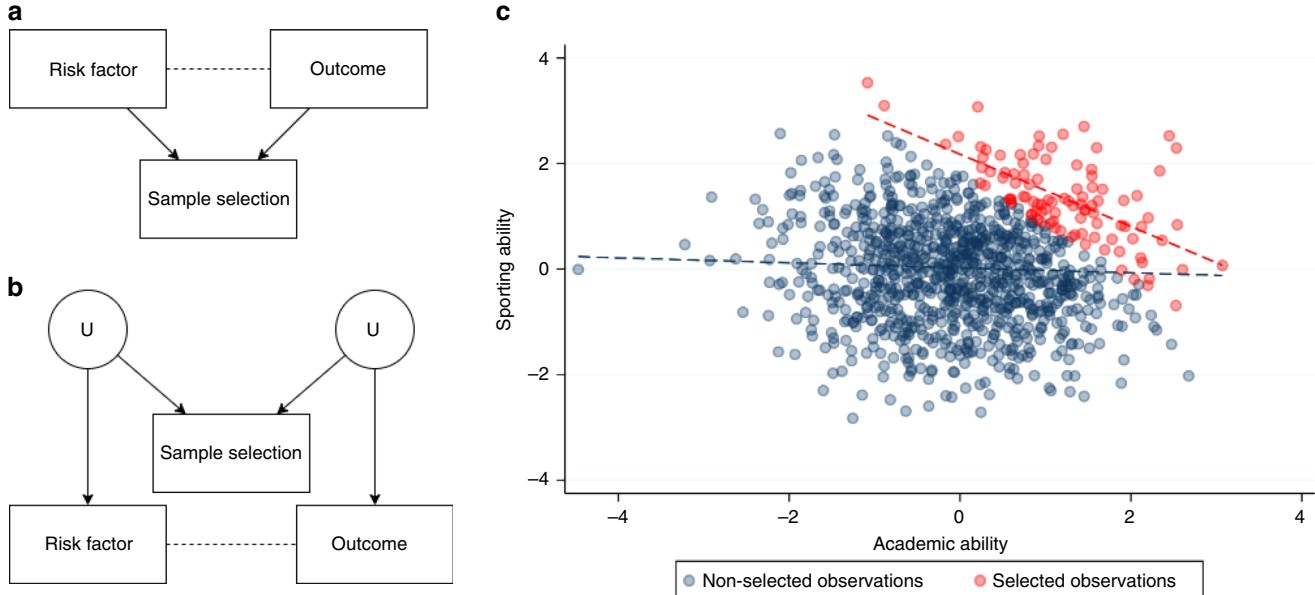

**Fig. 1 Illustrative example of collider bias. a** A directed acyclic graph (DAG) illustrating a scenario in which collider bias would distort the estimate of the causal effect of the risk factor on the outcome. Directed arrows indicate causal effects and dotted lines indicate induced associations. Note that the risk factor and the outcome can be associated with sample selection indirectly (e.g. through unmeasured confounding variables), as shown in **b**. The type of collider bias induced in graph (**b**) is sometimes referred to as M-bias. To illustrate the example in **a**, consider academic ability and sporting ability to each influence selection into a prestigious school. As shown in **c**, these traits are negligibly correlated in the general population (blue dotted line), but because they are selected for enrolment they become strongly correlated when analysing only the selected individuals (red dotted line).

this section of the population, health workers will therefore generally appear to have relatively low severity compared to others tested, inducing a negative association in our sample, which does not reflect the true relationship in the target population (Fig. 2b). It is clear that naive analysis using this selected sample will generate unreliable causal inference, and unreliable predictors to be applied to the general population.

In this paper, we discuss why collider bias should be of particular concern to observational studies of COVID-19 infection and disease risk, and show how sample selection can lead to dramatic biases. We then go on to describe the approaches that are available to explore and mitigate this problem.

## Results and discussion
**Why observational COVID-19 research is particularly susceptible to collider bias**. Though unquestionably valuable, observational datasets can be something of a black box because the associations estimated within them can be due to many different mechanisms. Consider the scenario in which we want to estimate the causal effect of a risk factor that is generalizable to a wider population such as the UK (the "target population"). Since we rarely observe the full target population, we must estimate this effect within a sample of individuals drawn from this population. If the sample is a true random selection from the population, then we say it is representative. Often, however, samples are chosen out of convenience or because the risk factor or outcome is only measured in certain groups (e.g. COVID-19 disease status is only observed for individuals who have received a test). Furthermore, individuals invited to participate in a sample may refuse or subsequently drop out. If characteristics related to sample inclusion also relate to the risk factor and outcome of interest, then this introduces the possibility of collider bias in our analysis.

Collider bias does not only occur at the point of sampling. It can also be introduced by statistical modelling choices. For example, whether it is appropriate to adjust for covariates in observational associations depends on where the covariates sit on

the causal pathway and their role in the data generating process[18–21]. If we assume that a given covariate influences both the hypothesised risk factor and the outcome (a confounder), it is appropriate to condition on that covariate to remove bias induced by the confounding structure. However, if the covariate is a consequence of either or both the exposure and the outcome (a collider), rather than a common cause (a confounder), then conditioning on the covariate can induce, rather than reduce, bias[22–24]. That is, collider bias can also be introduced when making statistical adjustments for variables that lie on the causal pathway between risk factor and outcome. A priori knowledge of the underlying causal structure of variables and whether they function as a common cause or common consequence of risk factor and outcome in the data generating process can be hard to infer. Therefore, it is appropriate to treat collider bias with a similar level of caution to confounding bias. We address ways of doing so later in this paper ("Methods for detecting and minimising the effects of collider bias").

There are multiple ways in which data are being collected on COVID-19 that can introduce unintentional conditioning in the selected sample. The characteristics of participants recruited are related to a range of factors including policy decisions, cost limitations, technological access, and testing methods. It is also widely acknowledged that the true prevalence of the disease in the population remains unknown[25]. Here we describe the forms of data collection for COVID-19 before detailing the circumstances surrounding COVID-19 that make its analysis susceptible to collider bias.

**COVID-19 sampling strategies and case/control definitions**. *Sampling conditional on voluntary participation (Case definition: probable COVID-19, Control definition: voluntary participant not reporting COVID-19 symptoms*, Fig. 2a): Probable COVID-19 status can be determined through studies that require voluntary participation. These may include, for example, surveys conducted by existing cohort and longitudinal studies[26,27], data linkage to

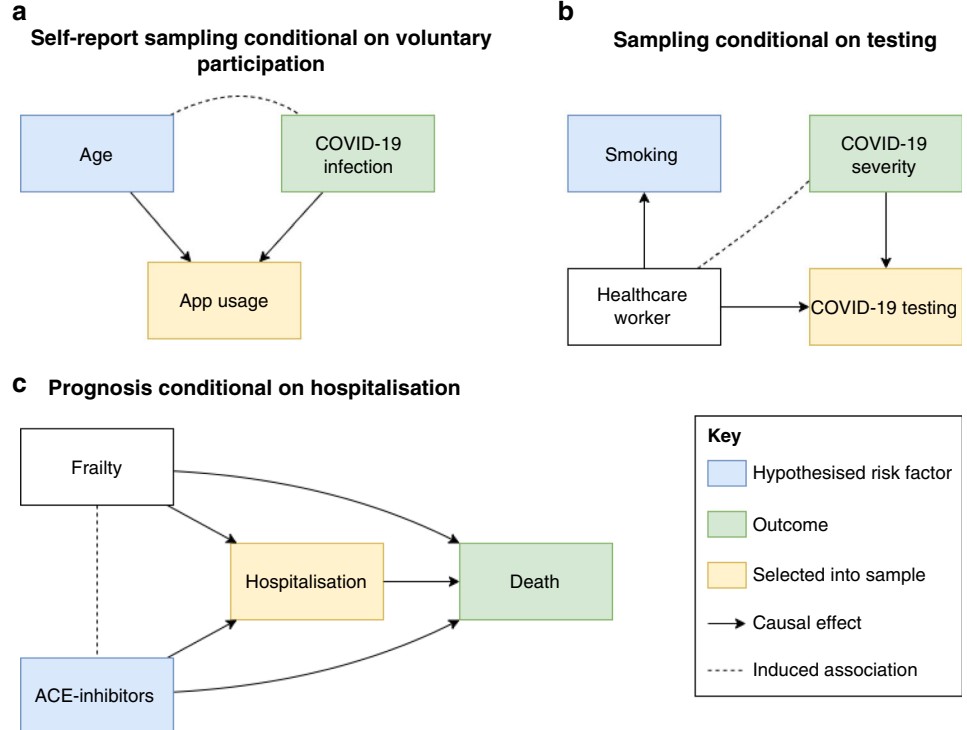

**Fig. 2 Collider bias induced by conditioning on a collider in three scenarios relating to COVID-19 analysis.** These are simplified Directed Acyclic Diagrams where only the main variables of interest have been represented for sake of illustrating collider bias scenarios. All assume no unspecified confounding or other biases. Rectangles represent observed variables and solid directed arrows represent causal effects. The dashed line represents an induced association when conditioning on the collider, which in these scenarios are variables that indicate whether an individual is selected into the sample. **a** When some hypothesised risk factor (e.g. age) and outcome (e.g. COVID-19 infection) each associate with sample selection (e.g. voluntary data collection via mobile phone apps), the hypothesised risk factor and outcome will be associated within the sample. The presence and direction of these biases are model dependent; where causes are supra-multiplicative they will be positively associated in the sample; where they are sub-multiplicative they will be negatively correlated, and where they are exactly multiplicative they will remain unassociated. We extend this scenario in **b** where the association between the hypothesised risk factor and the collider does not need to be causal. **c** When inferring the influence of some hypothesised risk factor on mortality, in an unselected sample the risk factor for infection is a causal factor for death (mediated by COVID-19 infection). However, if analysed only amongst individuals who are known to have COVID-19 (i.e. we condition on the COVID-19 infection variable) then the risk factor for infection will appear to be associated with any other variable that influences both infection and progression. In many circumstances, this can lead to a risk factor for disease onset that appears to be protective for disease progression. Each of these scenarios represents those described in the main text.

administrative records available in some cohort studies such as the UK Biobank[28], or mobile phone-based app programmes[29,30]. Participation in scientific studies has been shown to be strongly non-random (e.g. participants are disproportionately likely to be highly educated, health conscious, and non-smokers), so the volunteers in these samples are likely to differ substantially from the target population[31–33]. See Box 2 and Fig. 3 for a vignette on how one study[30] explored collider bias in this context.

**Sample selection pressures for COVID-19 studies**. We can stratify the sampling strategies above into three primary sampling frames. The first of these frames is sampling based on voluntary participation, which is inherently non-random due to the factors that influence participation. The second of these is sampling frames using COVID-19 testing results. With few notable exceptions (e.g. refs. [3,34]), population testing for COVID-19 is not generally performed in random samples. The third of these frames is sampling based on hospitalised patients, with or without COVID-19. This is again, necessarily non-random as it conditions on hospital admission.

Box 3 and Fig. 3 illustrate the breadth of factors that can induce sample selection pressure. While some of the factors that impact the sampling processes may be common across all modes of sampling listed above, some will be mode specific. These factors

will likely differ in how they operate across national and healthcare system contexts. Here we list a series of possible selection pressures and how they impact different COVID-19 sampling frames. We also describe case identification/definition and detail how they may bias inference if left unexplored.

*Symptom severity*: This will conceivably bias all three major sampling frames, although is most simply understood in context of testing. Several countries adopted the strategy of offering tests predominantly to patients experiencing symptoms severe enough to require medical attention, e.g. hospitalisation, as was the case in the UK until the end of April 2020. Many true positive cases in the population will therefore remain undetected and will be less likely to form part of the sample if enrolment is dependent upon test status. High rates of asymptomatic virus carriers or cases with the atypical presentation will further compound this issue.

*Symptom recognition*: This will also bias all three sampling frames as entry into all samples is conditional on symptom recognition. Related to but distinct from symptom severity, COVID-19 testing will vary based upon symptom recognition[35]. If an individual fails to recognise the correct symptoms or deems their symptoms to be nonsevere, they may simply be instructed to self-isolate and not receive a COVID-19 test. Individuals will assess their symptom severity differently; those with health-related anxiety may be more likely to over-report symptoms,

---

**Box 2 | The potential association between ACE inhibitors and COVID-19: why sampling bias matters**

One research question that has gained attention is whether blood pressure-lowering drugs, such as ACE inhibitors (ACE-i) and angiotensin-receptor blockers (ARBs), which act on the Renin–Angiotensin–Aldosterone System (RAAS) system, make patients more susceptible to COVID-19 infection[83–87]. Relationships between ACE-i/ARBs and COVID-19 are to be investigated in clinical trials[88,89], but in the meantime have been rapidly investigated through observational studies[90–92]. One such recent analysis used data from a UK COVID-19 symptom tracker app[93], which was released in March just before the UK Lockdown policy was implemented to increase social distancing. The app allows members of the public to contribute to research through self-reporting data including demographics, conditions, medications, symptoms and COVID-19 test results. The researchers observed that people reporting ACE-i use were twice as likely to self-report COVID-19 infection based on symptoms, even after adjusting for differences in age, BMI, sex, diabetes, and heart disease[30]. This association was attenuated from an odds ratio greater than four from an earlier freeze of the data, which comprised a smaller (and likely more selected) sample. When estimating the association only amongst individuals tested for COVID-19 infection the direction of the effect reversed, and ACE-i use appeared mildly protective. The simplest explanation for such volatility in estimates is that sample selection differed across the sample subsets.

The researchers diligently investigated the role of collider bias by performing parameter search sensitivity analyses (see below), finding that non-random sampling was sufficient to explain the association. If taking ACE-i and having COVID-19 symptoms would lead to being either less or more likely to sign up to the app or contribute data, this could induce an association between these factors (Fig. 2a). Since ACE-is are prescribed to those with diabetes, heart disease, or hypertension, ACE-i users are likely to be considered high-risk for COVID-19[94]. They are therefore potentially more sensitised to their current health status and may be more likely to use the app[95,96]. People who are COVID-19 symptomatic may also be more likely to remember to contribute data than asymptomatic people. Taken together, this could result in a false or inflated association between taking ACE-i and COVID-19. However, in reality, deciding in which direction ACE-i and COVID-19 symptoms would influence participation is complicated. For example, people with severe COVID-19 symptoms who are hospitalised could be too ill to contribute data.

Careful consideration is required for each set of exposures and outcomes that are studied. Amongst those participants who were actually tested in the COVID-19 symptom tracker app study, there was no evidence for an association between ACE-i use and COVID-19 positive status[30]. In this analysis there are joint selection pressures of factors underlying a) being tested and b) app participation.

Should ACE-i use truly increase risk of COVID-19 infection, it could imply that observational results for disease progression studies are influenced by collider bias. For example, it has been reported that ACE-i/ARB use may be protective against severe symptoms, conditional on already being infected[97,98], which is consistent with index event bias as illustrated in Fig. 2c.

It is important to consider the plausibility of the different selection pathways, both statistically (for example, through methods such as bounds and parameter searches) and biologically. Such considerations will ensure that data interpretation is at least robust to known biases of unknown magnitude, and that policy decisions are based on the best interpretation of the scientific evidence. Indeed, in consideration of the benefits that ACE-i/ARBs have on the cardio-respiratory system, current guidelines have continued to recommend the use of these drugs until there is sufficiently reliable scientific evidence against this[99,100].

*Sampling conditional on being tested for active COVID-19 infection (Case definition: positive test for COVID-19, Control definition: negative test for COVID-19, Fig. 2b):* PCR antigen tests are used to confirm a suspected (currently active) COVID-19 infection. Studies that aim to determine the risk factors for confirmed current COVID-19 infection therefore rely on participants having received a COVID-19 antigen test (hereafter for simplicity: COVID-19 test or test). Unless a random sample or the entire population are tested, these studies are liable to provide a biased estimate of active COVID-19 infection prevalence in the general population. The resource for testing is limited, so different countries have been using different (pragmatic) strategies for prioritising testing, including on the basis of characteristics such as occupation, symptom presentation and perceived risk. See Box 3 for an investigation into whether testing is non-random with respect to a range of measurable potential risk factors, using the recently released COVID-19 test data in the UK-Biobank.

---

while those with less information on the pandemic or access to health advice may be under-represented. This will functionally act as a differential rate of false-negatives across individuals based on symptom recognition, which could be consequential in giving the high estimates of asymptomatic cases and transmission[36]. Changing symptom guidelines is likely to compound this problem, which could induce systematic relationships between symptom presentation and testing[35,37]. Here, groups with lower awareness (for example, due to inadequate public messaging or language barriers) may have higher thresholds for getting tested, and therefore those who test positive will appear to have greater risk of severe COVID-19 outcomes.

*Occupation*: Exposure to COVID-19 is patterned with respect to occupation. In many countries, frontline healthcare workers are far more likely to be tested for COVID-19 than the general population[5,38] due to their proximity to the virus and the potential consequences of infection-related transmission[39]. As such, they will be heavily over-represented in samples conditional on test status. Other key workers may be at high risk of infection due to large numbers of contacts relative to non-key workers, and may therefore be over-represented in samples conditional on positive test status or COVID-related death. Any factors related to these occupations (e.g. ethnicity, socio-economic position, age and baseline health) will therefore also be associated with sample selection. Figure 2b illustrates an example where the hypothesised risk factor (smoking) does not need to influence sample selection (hospitalised patients) causally, it could simply be associated due to confounding between the risk factor and sample selection (being a healthcare worker).

*Ethnicity*: Ethnic minorities are also more likely to be infected with COVID-19[40]. Adverse COVID-19 outcomes are considerably worse for individuals of some ethnic minorities[41]. This could conceivably bias estimated associations within sampling frames based within hospitalised patients, as in many countries, ethnic minority groups are over-represented as ethnic inequalities in health are pervasive and well-documented. Furthermore, ethnic minority groups are more likely to be key workers, who are more likely to be exposed to COVID-19[42]. Cultural environment (including systemic racism) and language barriers may negatively affect entry into studies, both based on testing and voluntary participation[43]. Ethnic minority groups may be more difficult to recruit into studies, even within a given area[44], and may affect the representativeness of the sample. Ethnic minorities were less likely to report being tested in our analysis of the UK Biobank data, where one of the strongest factors associated with being tested was the first genetic principal component, which is a marker for ancestry (Box 3). Thus, this could present as above, with ethnic minorities' presentation to medical care being conditional on more severe symptoms.

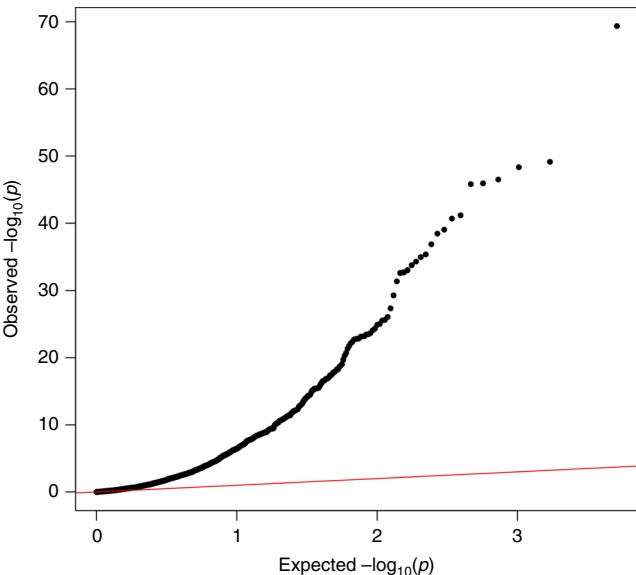

**Fig. 3 Quantile-Quantile plot of −log10 p-values for factors influencing being tested for COVID-19 in UK Biobank.** The x-axis represents the expected p-value for 2556 hypothesis tests and y-axis represents the observed p-values. The red line represents the expected relationship under the null hypothesis of no associations.

*Internet access and technological engagement*: This will primarily bias voluntary recruitment via apps, although may also be associated with increased awareness and bias testing via the symptom recognition pathway. Sample recruitment via internet applications is known to under-represent certain groups[32,51]. Furthermore, this varies by sampling design, where voluntary or "pull-in" data collection methods have been shown to produce more engaged but less representative samples than advertisement based or "push out" methods[33]. These more engaged groups likely have greater access to electronic methods of data collection, and greater engagement in social media campaigns that are designed to recruit participants. As such, younger people are more likely to be over-represented in app-based voluntary participation studies[29].

*Medical and scientific interest*: Studies recruiting voluntary samples may be biased as they are likely to contain a disproportionate amount of people who have a strong medical or scientific interest. It is likely that these people will themselves have greater health awareness, healthier behaviour, be more educated, and have higher incomes[31,52].

Many of the factors for being tested or being included in datasets described here are borne out in the analysis of the UK Biobank test data (Box 3). The key message is that when sample recruitment is non-random, there is an incredibly broad range of ways in which that non-randomness can undermine study results.

**Methods for detecting and minimising the effects of collider bias.** In this section, we describe methods to either address collider bias or evaluate the sensitivity of results to collider bias. As with confounding bias, it is generally not possible to prove that any of the methods has overcome collider bias. Therefore, sensitivity analyses are crucial in examining the robustness of conclusions to plausible selection mechanisms[18,19].

A simple, descriptive technique to evaluate the likelihood and extent of collider bias induced by sample selection is to compare means, variances and distributions of variables in the sample with those in the target population (or a representative sample of the target population)[16]. This provides information about the profile of individuals selected into the sample from the target population of interest, such as whether they tend to be older or more likely to have comorbidities. It is particularly valuable to report these comparisons for key variables in the analysis, such as the hypothesised risk factor and outcome, and other variables related to these. With respect to the analysis of COVID-19 disease risk, one major obstacle to this endeavour is that in most cases the actual prevalence of infection in the general population is unknown. While it is encouraging if the sample estimates match their population-level equivalents, it is important to recognise that this does not definitively prove the absence of collider bias[53]. This is because factors influencing selection could be unmeasured in the study, or factors interact to influence the selection and go undetected when comparing marginal distributions.

Each method's applicability crucially depends on the data that are available on non-participants. These methods can broadly be split into two categories based on the available data: nested and non-nested samples. A nested sample refers to the situation when key variables are only measured within a subset of an otherwise representative "super sample", thus forcing analysis to be restricted to this sub-sample. An example close to this definition is the sub-sample of individuals who have received a COVID-19 test nested within the UK Biobank cohort (though, it is clear that the UK Biobank cohort is itself non-randomly sampled[16]). For nested samples researchers can take advantage of the data available in the representative super-sample. A non-nested sample refers to the situation when data are only available in

*Frailty*: Defined here as greater susceptibility to adverse COVID-19 outcomes, frailty is more likely to be present in certain groups of the population, such as older adults in long-term care or assisted living facilities, those with pre-existing medical conditions, obese groups, and smokers. These factors are likely to strongly predict hospitalisation. At the same time, COVID-19 infection and severity likely have an influence on hospitalisation[8–10,45], meaning investigating these factors within hospitalised patients may induce collider bias. In addition, groups may be treated differently in terms of reporting on COVID-19 in different countries[46]. For example, in the UK early reports of deaths "due to COVID-19" may have been conflated with deaths "while infected with COVID-19"[47]. Individuals at high risk are more likely to be tested in general, but specific demographics at high risk such as those in long-term care or assisted living facilities have been less likely to be sampled by many studies[46]. Frailty also predicts hospitalisation differentially across different groups, for instance, an older individual with very severe COVID-19 symptoms in an assisted living facility may not be taken to hospital where a younger individual would[48].

*Place of residence and social connectedness*: A number of more distal or indirect influences on sample selection likely exist. People with better access to healthcare services may be more likely to be tested than those with poorer access. Those in areas with a greater number of medical services or better public transport may find it easier to access services for testing, while those in areas with less access to medical services may be more likely to be tested[49]. People living in areas with stronger spatial or social ties to existing outbreaks may also be more likely to be tested due to increased medical vigilance in those areas. Family and community support networks are also likely to influence access to medical care, for instance, those with caring responsibilities and weak support networks may be less able to seek medical attention[50]. Connectedness is perhaps most likely to bias testing sampling frames, as testing is conditional on awareness and access. However, it may also bias all three major sampling frames through a similar mechanism to symptom recognition.

**Box 3 | Factors influencing being tested in UK Biobank**

In April 2020, General Practices across the UK released primary care data on COVID-19 testing for linkage to the participants in the UK Biobank project[101,102] and results from analyses are already appearing[103]. Of the 486,967 participants, 1410 currently have data on COVID-19 testing. While it may be tempting to look for factors that influence whether an individual tests positive, it is crucial to evaluate the potential that those tested are not a random sample of the UK-Biobank participants (who are themselves not a random sample of the UK population).

We examined 2556 different characteristics for association with whether or not a UK Biobank participant had been tested for COVID-19. There was very large enrichment for associations (Fig. 3), with 811 of the phenotypes (32%) giving rise to a false discovery rate < 0.05. These associations involved a wide range of traits, including measures of frailty, medications used, genetic principal components, air pollution, socio-economic status, hypertension and other cardiovascular traits, anthropometric measures, psychological measures, behavioural traits, and nutritional measures. A full list of all traits assessed and their associations with whether a participant had COVID-19 test data is available in Source Data File. The first genetic principal component, which relates to global ancestral groups, was one of the strongest associations with being tested, which may have implications for interpreting ethnicity differences in COVID-19 test results[103].

We cannot know the actual COVID-19 prevalence amongst all participants, but if it is different from the prevalence amongst those tested, then every one of the traits listed above could be associated with COVID-19 in the dataset solely due to collider bias, or at least the magnitude of those associations could be biased as a result. The fact that the UK Biobank data are already a non-random sample of the UK population further complicates the matter[16].

Ideally, inverse-probability weighted regressions would be performed to minimise any such bias, as illustrated in the Supplementary Note. However, because we cannot know whether participants outside of the tested group had COVID-19 (i.e. the 'sampling fractions'), such weights cannot be calculated without strong assumptions that are currently untestable[59]. Inverse probability weighting also depends on the selection model being correctly specified, including that all characteristics predicting selection (that are related to variables in the analysis model) have been included, and in the right functional form. As with unmeasured confounding, there is always the possibility of having unmeasured selection factors.

*Sampling conditional on having a positive test for active COVID-19 infection (Case definition: severe COVID-19 symptoms, Control definition: positive COVID-19 test with mild symptoms, Fig. 2b):* Studies that aim to determine the risk factors for severity of confirmed current COVID-19 infection therefore rely on participants having received a COVID-19 test, and that the result of the test was positive. As above, testing is unlikely to be random, and conditioning on the positive result will also mean bias can be induced by all factors causing infection, as well as those causing increased likelihood of testing.

*Sampling conditional on hospitalisation (Case definition: hospitalised patients with COVID-19 infection, Control definition: hospitalised patients without COVID-19 infection):* An important source of data collection is from existing patients or hospital records. Several studies have emerged which make causal inference from such selected samples[8,9,45]. COVID-19 infection influences hospitalisation, as do a large number of other health conditions. By analysing only hospitalised samples, anything that influences hospitalisation will become negatively associated with COVID-19 infection (in the marginal case of no interactions).

*Sampling conditional on hospitalisation and having a positive test for active COVID-19 infection (Case definition: COVID-19 death, Control definition: non-fatal COVID-19 related hospitalisation, Fig. 2c):* Many studies have started analysing the influences on disease progression once individuals are infected, or infected and then admitted to hospital (i.e. the factors that influence survival). Such datasets necessarily condition upon a positive test. Figure 2c illustrates how this so-called "index event bias" is a special case of collider bias[16,104,105]. If we accept that COVID-19 increases mortality, and there are risk factors for infection of COVID-19, then in a representative sample of the target population, any cause of infection would also exert a causal influence on mortality, mediated by infection. However, once we condition on being infected, all factors for infection become correlated with each other. If some of those factors influence both infection and progression, then the association between a factor for infection and death in the selected sample will be biased. This could lead to factors that increase risk of infection falsely appearing to be protective for severe progression[1,106]. An example of this relevant to COVID-19 is discussed in Box 2.

---

an unrepresentative sample. An example of this is samples of hospitalized individuals, in which no data are available on non-hospitalized individuals. It is typically more challenging to address collider bias in non-nested samples. A guided analysis illustrating both types of sensitivity analyses using UK Biobank data on COVID-19 testing is presented in Supplementary Note 1.

*Nested samples*: Inverse probability weighting is a powerful and flexible approach to adjust for collider bias in nested samples[54,55]. The causal effect of the risk factor on the outcome is estimated using weighted regression, such that participants who are overrepresented in the sub-sample are down-weighted and participants who are underrepresented are up-weighted. In practice, we construct these weights by estimating the likelihood of different individuals being selected into the sample from the representative super-sample based on their measured covariates[56]. For example, we could use data from the full UK Biobank sample to estimate the likelihood of individuals receiving a test for COVID-19 and use these weights in analyses that have to be restricted to the sub-sample of tested individuals (e.g. identifying risk factors for testing positive). Seaman and White provide a detailed overview of the practical considerations and assumptions for inverse probability weighting, such as correct specification of the "sample selection model" (a statistical model of the relationship between measured covariates and selection into the sample, used to construct these weights), variable selection and

approaches for handling unstable weights (i.e. weights which are zero or near-zero).

An additional assumption for inverse probability weighting is that each individual in the target population must have a non-zero probability of being selected into the sample. Neither this assumption, nor the assumption that the selection model has been correctly specified, are testable using the observed data alone. A conceptually related approach, using propensity score matching, is sometimes used to avoid index event bias[57,58]. There also exist sensitivity analyses for misspecification of probability weights. For example, Zhao et al. develop a sensitivity analysis for the degree to which estimated probability weights differ from the true unobserved weights[59]. This approach is particularly useful when we can estimate probability weights including some, but not necessarily all, of the relevant predictors of sample inclusion. For example, we could estimate weights for the likelihood of receiving a COVID-19 test among UK Biobank participants, however, we are missing key predictors such as symptom presentation and measures of healthcare-seeking behaviour.

*Non-nested samples*: When we only have data on the study sample (e.g. only data on participants who were tested for COVID-19) it is not possible to estimate the selection model directly since non-selected (untested) individuals are unobserved. Instead, it is important to apply sensitivity analyses to assess the plausibility that sample selection induces collider bias.

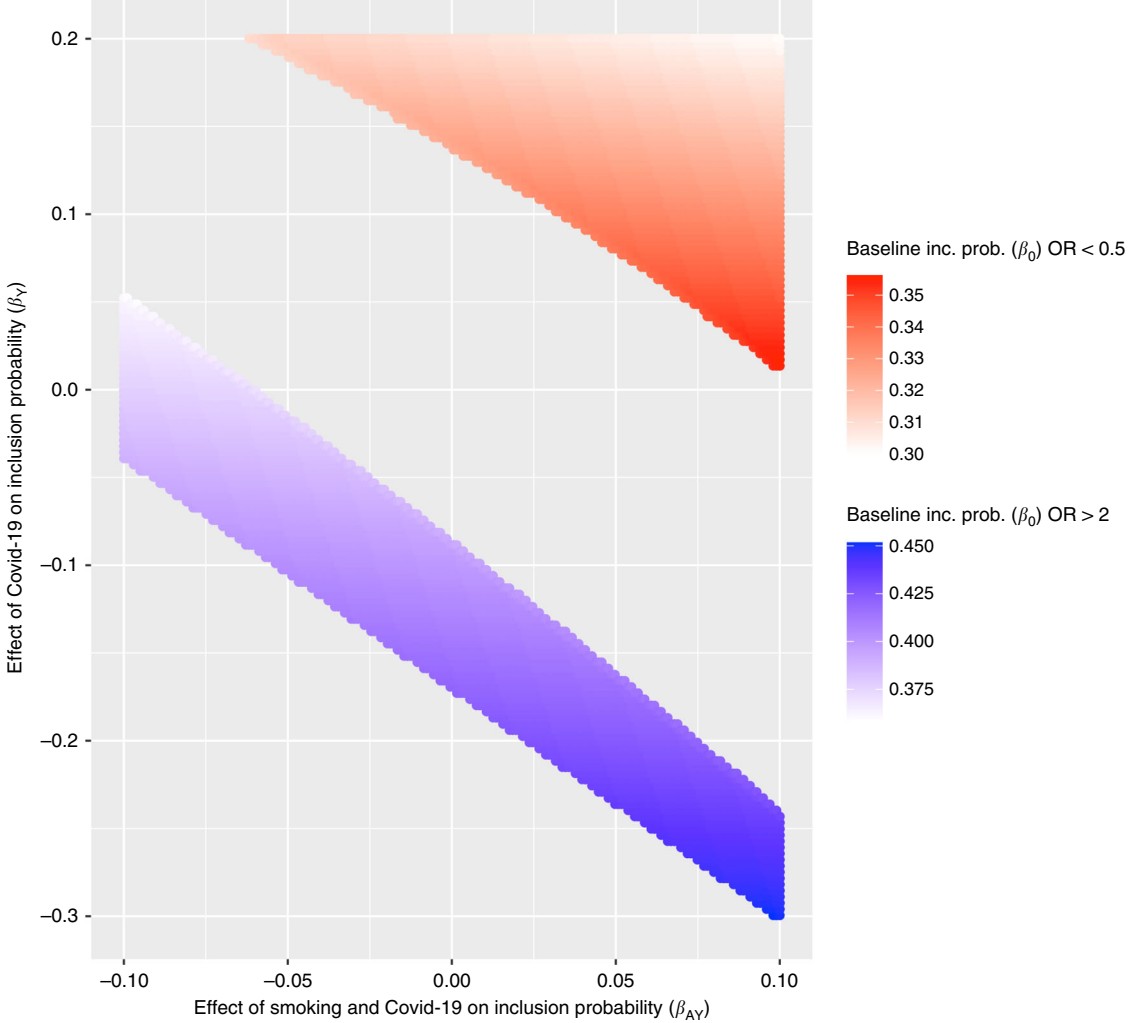

**Fig. 4 Example of large associations induced by collider bias under the null hypothesis of no causal relationship, using scenarios similar to those reported for the observed protective association of smoking on COVID-19 infection.** Assume a simple scenario in which the hypothesised exposure (A) and outcome (Y) are both binary and each influence probability of being selected into the sample (S) e.g. $P(S = 1|A, Y) = \beta_0 + \beta_A + \beta_Y + \beta_{AY}$ where $\beta_0$ is the baseline probability of being selected, $\beta_A$ is the effect of A, $\beta_Y$ is the effect of Y and $\beta_{AY}$ is the effect of the interaction between A and Y. The selection mechanism in question is represented in Fig. 1b (without the interaction term drawn). This plot shows which combinations of these parameters would be required to induce an apparent risk effect with magnitude OR > 2 (blue region) or an apparent protective effect with magnitude OR < 0.5 (red region) under the null hypothesis of no causal effect[61]. To create a simplified scenario similar to that in Miyara et al. we use a general population prevalence of smoking of 0.27 and a sample prevalence of 0.05, thus fixing $\beta_A$ at 0.22. Because the prevalence of COVID-19 is not known in the general population, we allow the sample to be over- or under-representative (y-axis). We also allow modest interaction effects. Calculating over this parameter space, 40% of all possible combinations lead to an artefactual 2-fold protective or risk association operating through this simple model of bias alone. It is important to disclose this level of uncertainty when publishing observational estimates.

*Bounds and parameter searches*: It is possible to infer the extent of collider bias given knowledge of the likely size and direction of influences of risk factor and outcome on sample selection (whether these are direct, or via other factors)[19,60,61]. However, this approach depends on the size and direction being correct, and there being no other factors influencing selection. It is therefore important to explore different possible sample selection mechanisms and examine their impact on study conclusions. We created a simple web application guided by these assumptions to allow researchers to explore simple patterns of selection that would be required to induce an observational association: http://apps.mrcieu.ac.uk/ascrtain/. In Fig. 4 we use a recent report of a protective association of smoking on COVID-19 infection[45] to explore the magnitude of collider bias that can be induced due to selected sampling, under the null hypothesis of no causal effect.

Several other approaches have also been implemented into convenient online web apps ("Appendix"). For example, Smith and VanderWeele proposed a sensitivity analysis which allows researchers to bound their estimates by specifying sensitivity parameters representing the strength of sample selection (in terms of relative risk ratios). They also provide an "E-value", which is the smallest magnitude of these parameters that would explain away an observed association[62]. Aronow and Lee proposed a sensitivity analysis for sample averages based on inverse probability weighting in non-nested samples where the weights cannot be estimated but are assumed to be bounded between two researcher-specified values[63]. This work has been generalised to regression models, also allowing relevant external information on the target population (e.g. summary statistics from the census) to be incorporated[64]. These sensitivity analysis approaches allow researchers to explore whether there are

credible collider structures that could explain away observational associations. However, they do not represent an exhaustive set of models that could give rise to bias, nor do they necessarily prove whether collider bias influences the results. If the risk factor for selection is itself the result of further upstream causes then it is important that the impact of these upstream selection effects are considered (i.e. not only how the risk factor influences selection but also how the causes of the risk factor and/or the causes of the outcome influence selection e.g. Fig. 2b). While these upstream causes may individually have a small effect on selection, it is possible that lots of factors with individually small effects could jointly have a large selection effect and introduce collider bias[65].

*Negative control analyses*: If there are factors measured in the selected sample that are known to have no influence on the outcome, then testing these factors for association with the outcome within the selected sample can serve as a negative control[66,67]. By virtue, negative control associations should be null, and they are therefore useful as a tool to provide evidence in support of selection. If we observe associations with larger magnitudes than expected then this indicates that the sample is selected on both the negative control and the outcome of interest[68,69].

*Correlation analyses*: Conceptually similar to the negative controls approach above, when a sample is selected, all the features that influenced selection become correlated within the sample (except for the highly unlikely case that causes are perfectly multiplicative). Testing for correlations amongst hypothesised risk factors where it is expected that there should be no relationship can indicate the presence and magnitude of sampling selection, and therefore the likelihood of collider bias distorting the primary analysis[70].

**Implications**. The majority of scientific evidence informing policy and clinical decision making during the COVID-19 pandemic has come from observational studies[71]. We have illustrated how these observational studies are particularly susceptible to non-random sampling. Randomised clinical trials will provide experimental evidence for treatment, but experimental studies of infection will not be possible for ethical reasons. The impact of collider bias on inferences from observational studies could be considerable, not only for disease transmission modelling[72,73], but also for causal inference[7] and prediction modelling[2].

While many approaches exist that attempt to ameliorate the problem of collider bias, they rely on unprovable assumptions. It is difficult to know the extent of sample selection, and even if that were known it cannot be proven that it has been fully accounted for by any method. Representative population surveys[34] or sampling strategies that avoid the problems of collider bias[74] are urgently required to provide reliable evidence. Results from samples that are likely not representative of the target population should be treated with caution by scientists and policy makers.

## Methods

**Factors influencing testing in the UK Biobank**. UK-Biobank phenotypes were processed using the PHESANT pipeline[75] and filtered to include only quantitative traits or case-control traits that had at least 10,000 cases. In addition, sex, genotype chip and the first 40 genetic principal components were included for analysis (2556 traits in total). A detailed description of how all the variables were formatted in this analysis has been provided in Mitchell and colleagues[76]. A "tested" variable was generated that indicated whether an individual had been tested for COVID-19 or not within UK Biobank, and logistic regression was performed for each of the 2556 traits against the "tested" variable. Code is available here: https://github.com/explodecomputer/covid_ascertainment. This research was conducted using the UK Biobank Resource applications 8786 and 15,825, and complied with all relevant ethical regulations.

**Sensitivity analysis of the effect of smoking on COVID-19 infection**. Given knowledge of an observational association estimate between exposure A and

outcome Y, here our objective is to estimate the extent to which A and Y must relate to sample selection in order to induce the reported observational association. Assume that the probability of being present in the sample, $P(S = 1)$ is a function of A and Y:

$$P(S = 1|A, Y) = \beta_0 + \beta_A A + \beta_Y Y + \beta_{AY} AY$$

Where $\beta_0$ is the baseline probability of any individual to be a part of our sample, $\beta_A$ is the differential probability of being sampled for individuals in the exposed group ($A = 1$), $\beta_Y$ is the differential probability of being sampled for cases ($Y=1$), and $\beta_{AY}$ is the differential probability of being sampled for cases in the exposed group ($A = 1, Y = 1$). Given this, we may derive the expected odds ratio in the selected sample under the null hypothesis of no association in the unselected sample[61]:

$$E\left[\widehat{OR_{S=1}}\right] = \frac{\beta_0(\beta_0 + \beta_A + \beta_Y + \beta_{AY})}{(\beta_0 + \beta_A)(\beta_0 + \beta_Y)}$$

To create a simplified scenario similar to that in Miyara et al. we use a general population prevalence of smoking of 0.27 and a sample prevalence of 0.05, thus fixing $\beta_A$ at 0.22. We then explore the values of $\beta_0$, $\beta_Y$ and $\beta_{AY}$ that would lead to $E\left[\widehat{OR_{S=1}}\right]>2$ or $E\left[\widehat{OR_{S=1}}\right]<0.5$ ... Analyses were performed using the AscRtain R package.

**Additional methods**. A reproducible guided analysis for performing several of the adjustment and sensitivity methods described in this paper is provided in the Supplementary Note. The Supplementary Note is also available as a living document here: https://mrcieu.github.io/ukbb-covid-collider/

Exploring bounds and spaces that could explain an observational association can be achieved using a range of packages and apps:

- AscRtain app: http://apps.mrcieu.ac.uk/ascrtain/
- CollideR app[15]: https://watzilei.com/shiny/collider/
- Selection bias app[62]: https://selection-bias.herokuapp.com/
- Bias app[61]: https://remlapmot.shinyapps.io/bias-app/
- Lavaan R package[77]: http://lavaan.ugent.be/
- Dagitty R package[78]: http://www.dagitty.net/
- simMixedDAG: https://github.com/IyarLin/simMixedDAG

**Reporting summary**. Further information on research design is available in the Nature Research Reporting Summary linked to this article.

## Data availability

All data analysed was provided by the UK Biobank and can be accessed via https://www.ukbiobank.ac.uk/. A detailed description of how the phenotype data analysed here was accessed and formatted is provided here: https://doi.org/10.5523/bris.pnoat8cxo0u52p6ynfaekeigi. Association results for each of 2556 variables in the UK Biobank cohort, testing for their influence on being tested for COVID-19. Source data are provided with this paper.

## Code availability

All code is available in the following github repositories:
https://github.com/MRCIEU/ukbb-covid-collider
https://github.com/explodecomputer/covid_ascertainment
https://github.com/explodecomputer/ascrtain
https://github.com/MRCIEU/PHESANT

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

## Acknowledgements

We are grateful to Josephine Walker for helpful comments on this manuscript. This research has been conducted using the UK Biobank Resource under Application Number 16729. The Medical Research Council (MRC) and the University of Bristol support the MRC Integrative Epidemiology Unit [MC_UU_00011/1, MC_UU_00011/3]. G.J.G. is supported by an ESRC postdoctoral fellowship [ES/T009101/1]. N.M.D. is supported by a Norwegian Research Council Grant number 295989. G.H. is supported by the Wellcome Trust and Royal Society [208806/Z/17/Z]. M.J.T. is supported by a Wellcome Trust studentship [220067/Z/20/Z]. AH is supported by an MRC grant [MR/S002634/1].

## Author contributions

G.H., N.M.D., L.Z. conceived the idea. G.H. and M.J.T. performed the analysis. G.J.G., G.H. and T.M.P. wrote the software. G.J.G., T.T.M., M.J.T., A.H., G.M., L.Z., N.M.D., G.H. wrote the paper. G.J.G., T.T.M., M.J.T., A.H., G.M., L.P., G.C.S., J.S., T.M.P., G.D.S., K.T., L.Z., N.M.D., G.H. discussed the results and contributed to the final paper.

## Competing interests

The authors declare no competing interests.
