## [Peer Review File · Nature Communications]

REVIEWER COMMENTS

Reviewer #1 (Remarks to the Author):

This paper brings up the important problem of collider-stratification bias, which I agree is being neglected in much COVID-19 research. In that sense, the paper has an important message to convey, not only for the COVID-19 situation, but also beyond. I found the paper generally well written, with useful links to the literature on sensitivity analyses; some of the boxes give interesting insight into the extent to which COVID-19 testing may be selective, for instance.

I believe that the paper could have been more convincing if the following points were addressed:

1. The paper should give more specific insight and empirical evidence for distorted associations. It now basically suggests that collider-stratification bias might be a problem, but readers will only be convinced if they see how big the bias could be. This could either be done by viewing a given sample as representative, for instance (even if it is not), and further selecting a subsample from it to then show how associations in that subsample are distorted relative to the full sample. It could also be done along the lines of the second point below.
2. the authors now suggest different ways of dealing with collider-stratification bias, but none of them is really applied to the COVID-19 setting. This makes it unconvincing, leaving the question whether perhaps these methods are too complicated to be useful. I strongly encourage the authors to consider negative control experiments, correlation analyses, and a sensitivity analysis along the lines of their web application, in a substantive COVID-19 application. If the authors are not effectively using these methods, then surely the readers will not.
3. there is no discussion of the methodology behind the web application. It would be important to understand what assumptions this application is making, and what are its limitations. The description of the application on the web is also too vague for it to be useful. For instance, β_A is defined as 'the effect on selection into the sample given $A=1$ is true', but this only carries meaning if it is clear on what scale the effect is defined. Also for this, a clear demonstration and explanation is essential, as well as guidance how the developed approach differs from the alternatives that are considered.

Reviewer #2 (Remarks to the Author):

Overall comments:

The authors of this manuscript focused on collider bias in COVID-19 research have written a timely manuscript on what I think is an important topic that has received too little attention in the literature.

However, with that said, I think there is significant need for improvement before this manuscript is suitable for publication. This is an article on a very specific topic, yet the writing is unfocused in places. Part of it is written as a general tutorial on collider bias (the introduction in particular). Collider bias has been thoroughly reviewed in the epidemiologic literature, and while I suspect many readers of this journal will not necessarily be familiar with it, the entire Introduction could really be condensed into two paragraphs- the current first paragraph and a second paragraph clearly describing collider bias. The two examples and Box 1 detract from the purpose of the article (to explain COVID related collider bias).

I particularly liked the section on COVID-19 sampling strategies and case definitions. It's without a doubt the strongest part of the manuscript in my opinion. I think the subsequent sections on "Sample Selection Pressures" should be integrated into the section on sampling strategies. For example, include the discussion of symptom severity under the heading "sampling conditional on

being tested for active COVID-19 infection". It helps to connect the dots for the reader of how the sample selection mechanism causes bias.

An important overall note about the manuscript and scientific writing. Please avoid the use of colloquial terms and be careful with the language you use in writing. For example, starting a paragraph with "Suppose..." is very informal as is using a phrase "unfortunate truth". Older adult is a preferred term compared to elder, and there are more accurate phrases than 'care home' (i.e. long term care facility, assisted living facility). These may seem like minor suggestions but overall, the informal writing detracts from the manuscript being reviewed.

Specific Comments:

Abstract- Need to include sentence defining collider bias since this won't be familiar to many readers of the journal who are non-epidemiologists

-I found Box 1 confusing and hard to follow. It distracted me from the purpose of what the authors are trying to describe in the introduction paragraph. Suggest moving to another section where it fits better, integrating the information into the text, or eliminating

-I agree the confounding receives the majority of attention as a bias, but is it being widely addressed and well understood in the context of COVID-19?

-Collider bias will not be familiar to many readers. I would suggest including a more thorough description of the bias at the beginning of the second paragraph. Also, how does this differ from selection bias (a concept that is more likely to be familiar to readers)

-In Figure 1A, I would suggest making the DAG specifically related to the causal question being answered in this article. So make Risk Factor and Outcome something related to COVID and sample selection could be replaced by hospitalization as described in the text. Actually, I see this is done in figure

2. Consider eliminating Figure 1 and describing Figure 2 in its place.

-References required for the second-last sentence of the second paragraph (The magnitude of this distortion...)

-Paragraphs 2,3,4 all start with "Suppose...", a very colloquial term.

-There are already numerous well-known articles describing collider bias in the epidemiologic literature. I don't think Example 1 is necessary in this manuscript. It's not related to the topic being discussed. Would either revise to make it a COVID-specific example, or, remove.

-Where is the discussion of figure 1B?

-In Example 2, why would the target population be all adults in a study of healthcare workers?

-“To give context on how serious a problem collider bias can be, there is a continuing debate in the literature about the extent to which it is appropriate to adjust for covariates in observational associations”. This sentence, and the ones that follow, confuse me. Adjusting for a confounder will not induce bias, adjusting for a collider will. The quoted sentence is not correct- there is minimal debate on this topic- adjusting for a confounder closes a 'backdoor path' therefore removing bias, whereas conditioning on a collider induces bias. Please clarify what debate you feel exists in the literature.

-What is the "hidden causal structure"?

-Box 3 and Figure 3 are unnecessary. Figure 3 should be eliminated. Recently, there has been

extensive discussion about the role (or lack thereof) of p-values in epidemiologic analyses. I understand the point the authors are trying to convey about Factors influencing being tested, and think that if they choose to retain Box 3 the focus should be on discussing the specific variables that were strongly associated with being tested, not p-values (or inverse probability weighting, for that matter).

-Frailty is not synonymous with individuals who live in assisted living facilities/long term care. Frailty is a distinct construct, which may be present in some older adults who live in assisted living, but not all, and may be present in community dwelling older adults.

-One of the most important aspects of this pandemic is racial/ethnic disparities in care, morbidity, and mortality. The token mention of 'Ethnicity' in the manuscript is inadequate to address this critically important issue.

-“The primary task in any analysis is to evaluate the extent to which sample selection is likely to have actually occurred”. I don't believe this is correct. Isn't the primary task of an analysis to evaluate the exposure-outcome relationship of interest?

-There are no references included in the first three paragraphs on “methods for overcoming bias”. This must be corrected.

-What are “subtle departures in the characteristics of the study sample”? How is this evaluated? What defines subtle?

-Is it ever possible to prove the absence of selection through validation?

-Are sensitivity analyses a method for overcoming collider bias? The title of the final section of the manuscript should be revised.

-Isn't it possible to use a form of inverse weighting (weighting by the inverse of selection probabilities) in non-nested samples? See Tim Lash's work on bias analysis for a thorough description.

-I understand the point the authors are trying to convey when describing a situation “where the entire dataset comprises only the selected samples used for hypothesis testing (stand-alone)”. However, I think the description provided may be confusing to the reader who is not as familiar with bias analysis techniques. I think many people would think “don't we always use all our data available for analyses?”. Also, as described in the text, external information is required for the bounding approach so it's not completely stand alone. Perhaps focusing on Nested vs. Non-nested samples would be more clear.

-There is a Figure 4 described in the Text but I don't see it in the manuscript file.

Response to reviewers

We are very grateful to the reviewers for their time and expertise in helping improve this paper. Responses are in italics and indented.

Reviewer #1 (Remarks to the Author):

This paper brings up the important problem of collider-stratification bias, which I agree is being neglected in much COVID-19 research. In that sense, the paper has an important message to convey, not only for the COVID-19 situation, but also beyond. I found the paper generally well written, with useful links to the literature on sensitivity analyses; some of the boxes give interesting insight into the extent to which COVID-19 testing may be selective, for instance.

Thank you for the time you have taken to review this paper, the suggestions have been very valuable.

I believe that the paper could have been more convincing if the following points were addressed:

1. The paper should give more specific insight and empirical evidence for distorted associations. It now basically suggests that collider-stratification bias might be a problem, but readers will only be convinced if they see how big the bias could be. This could either be done by viewing a given sample as representative, for instance (even if it is not), and further selecting a subsample from it to then show how associations in that subsample are distorted relative to the full sample. It could also be done along the lines of the second point below.

Thank you for this suggestion, we agree that it is important for readers to see the magnitude of the problem. We have done two things, first demonstrated how volatile reported associations of ACE-i on COVID-19 are in different sample subsets of the same study; and second created a walkthrough analysis of UK-Biobank COVID-19 test data, showing the differences in simple associations in the tested subset compared to the entire sample. We also provide an illustrative example in Figure 1.

The final section of the introduction now reads:

"In this paper, we discuss why collider bias should be of particular concern to observational studies of COVID-19 infection and disease risk, and show how sample selection can lead to dramatic biases. We then go on to describe the approaches that are available to explore and mitigate this problem."

In our analysis of UKBiobank data, we show that the age-sex association in the tested subgroup is 7 times larger than in the total sample (Supplementary Note).

We demonstrate that the large association of smoking on covid-19 infection could easily be explained entirely by collider bias (Figure 4).

We demonstrate that large associations of ACE inhibitors on covid-19 infection drop from $OR > 4$ to $OR < 1$ depending on the time and type of sampling, and that any association can be easily explained by collider bias (Box 2).

We also show the extent of the factors influencing non-randomness of the testing sample in UK Biobank in Box 3.

2. The authors now suggest different ways of dealing with collider-stratification bias, but none of them is really applied to the COVID-19 setting. This makes it unconvincing, leaving the question whether perhaps these methods are too complicated to be useful. I strongly encourage the authors to consider negative control experiments, correlation analyses, and a sensitivity analysis along the lines of their web application, in a substantive COVID-19 application. If the authors are not effectively using these methods, then surely the readers will not.

This is a useful suggestion and we have now created a reproducible tutorial based on the UK-Biobank COVID-19 test data (Supplementary Note, available here: <https://mrcieu.github.io/ukbb-covid-collider/>). It illustrates a number of the methods described for adjusting for collider bias. In addition to this, we have provided a tutorial on how to reproduce the analyses in the AscRtain webapp, and we note that we are already linking to several pre-existing methods with their own implementations and tutorials. Overall, we hope that this paper will provide a useful practical resource for researchers.

3. there is no discussion of the methodology behind the web application. It would be important to understand what assumptions this application is making, and what are its limitations. The description of the application on the web is also too vague for it to be useful. For instance, beta_A is defined as `the effect on selection into the sample given

A=1 is true', but this only carries meaning if it is clear on what scale the effect is defined. Also for this, a clear demonstration and explanation is essential, as well as guidance how the developed approach differs from the alternatives that are considered.

The web-app has been updated substantially, and the interface and pedagogical elements improved. It now contains a far more comprehensive "Under the Hood" section alongside "Estimation", which details in greater depth the specification and interpretation of the parameters which are being selected by the user. There is also further signposting of the proofs that are the basis of the web-app. There is also a worked example which demonstrates use of the app using a simplified example taken from the early literature that emerged on smoking and COVID-19.

Reviewer #2 (Remarks to the Author):

Overall comments:

The authors of this manuscript focused on collider bias in COVID-19 research have written a timely manuscript on what I think is an important topic that has received too little attention in the literature. However, with that said, I think there is significant need for improvement before this manuscript is suitable for publication.

We are very grateful for the reviewer's time and expertise, the comments have greatly improved the manuscript.

This is an article on a very specific topic, yet the writing is unfocused in places. Part of it is written as a general tutorial on collider bias (the introduction in particular). Collider bias has been thoroughly reviewed in the epidemiologic literature, and while I suspect many readers of this journal will not necessarily be familiar with it, the entire Introduction could really be condensed into two paragraphs- the current first paragraph and a second paragraph clearly describing collider bias. The two examples and Box 1 detract from the purpose of the article (to explain COVID related collider bias).

We have now adapted our introduction to provide a clearer description of collider bias and provide better structure as suggested by the reviewer. We however wish to retain the COVID example and Box 1 as a means of explaining collider bias through both an accessible example and an applied example. As the reviewer points out, many readers may not be familiar with collider bias so we wish to make our manuscript as accessible as possible to increase awareness. The example in the introduction now reads:

“As illustration, consider the hypothesis that being a health worker is a risk factor for severe COVID-19 disease. Under the assumption of a higher viral load due to their occupational exposure, healthcare workers will on average experience more severe COVID-19 symptoms compared to the general population. The target population within which we wish to test this hypothesis is adults in any occupation (or unemployed); the exposure is being a health worker the outcome is COVID-19 symptom severity. The only way we can reliably estimate COVID-19 status and severity is by considering individuals who have a confirmed positive PCR test for COVID-19. However, restrictions on availability of testing especially in the early stages of the pandemic means that the available study sample is necessarily restricted to those individuals who have been tested for active COVID-19 infection. If we take the UK as an example (until late April 2020), let us assume a simplified scenario where all tests were performed either on frontline health workers (as critical vectors for disease among high risk individuals), or members of the general public who had symptoms severe enough to require hospitalisation (as high risk individuals). In this testing framework, our sample of participants will have been selected for both the hypothesised risk factor (being a healthcare worker) and the outcome of interest (severe symptoms). Our sample will therefore contain all health workers who are tested regardless of their symptom severity, while only non-health workers with severe symptoms will be included. In this section of the population, health workers will therefore generally appear to have relatively low severity compared to others tested, inducing a negative association in our sample, which does not reflect the true relationship in the target population (**Figure 2B**). It is clear that naive analysis using this selected sample will generate unreliable causal inference, and unreliable predictors to be applied to the general population.”

I particularly liked the section on COVID-19 sampling strategies and case definitions. It's without a doubt the strongest part of the manuscript in my opinion. I think the subsequent sections on “Sample Selection Pressures” should be integrated into the section on sampling strategies. For example, include the discussion of symptom severity under the heading “sampling conditional on being tested for active COVID-19 infection”. It helps to connect the dots for the reader of how the sample selection mechanism causes bias.

Thank you for this suggestion. We have now restructured the section on Sample selection pressure, explaining how each of the factors relates to each of the sampling frames.

An important overall note about the manuscript and scientific writing. Please avoid the use of colloquial terms and be careful with the language you use in writing. For example, starting a paragraph with “Suppose...” is very informal as is using a phrase “unfortunate truth”. Older adult is a preferred term compared to elder, and there are

more accurate phrases than 'care home' (i.e. long term care facility, assisted living facility). These may seem like minor suggestions but overall, the informal writing detracts from the manuscript being reviewed.

We have reviewed the text throughout the manuscript to avoid colloquial and informal wording. We have also edited all instances of "elderly" to "older adult" and all instances of "care homes" to "long term care or assisted living facility".

Specific Comments:

Abstract- Need to include sentence defining collider bias since this won't be familiar to many readers of the journal who are non-epidemiologists.

We have now included a brief definition of collider bias in the abstract, as follows: "Collider bias can be induced through sampling when two or more variables of interest influence the likelihood of an observation being sampled, distorting associations between these variables in the dataset."

-I found Box 1 confusing and hard to follow. It distracted me from the purpose of what the authors are trying to describe in the introduction paragraph. Suggest moving to another section where it fits better, integrating the information into the text, or eliminating

Thank you for bringing this to our attention. Upon re-reading box 1 we realise it is somewhat confusingly written. Our intention here is to provide some explanation for the use of the term 'risk factor' that we use throughout the paper, because it is somewhat ambiguous in meaning and without explanation some of the explanations in the main text could be misconstrued. This is largely because collider bias poses different problems to causal inference and prediction. We opted to put it in a box to avoid breaking the flow of the introduction, but feel it needs to be explained at first mention of the term 'risk factor'. We have now re-written Box 1 to improve clarity.

-I agree the confounding receives the majority of attention as a bias, but is it being widely addressed and well understood in the context of COVID-19?

Our opinion is that, in general, attempts are being made to address confounding in COVID-19 studies. There is of course considerable variation in how well studies are addressing confounding – a notable example of a study doing a poor job at appropriately identifying and controlling for confounding is that of male

baldness as 'risk factor' for COVID death, which didn't formally adjust for age! We have now included a caveat to this statement, (p. 2, "bar rare exceptions"), however we wish to keep the focus of our manuscript strongly centred on collider bias rather than confounding.

-Collider bias will not be familiar to many readers. I would suggest including a more thorough description of the bias at the beginning of the second paragraph. Also, how does this differ from selection bias (a concept that is more likely to be familiar to readers)

We have now adapted our introduction to provide a more thorough description of collider bias and provide better structure as suggested by the reviewer above.

-In Figure 1A, I would suggest making the DAG specifically related to the causal question being answered in this article. So make Risk Factor and Outcome something related to COVID and sample selection could be replaced by hospitalization as described in the text. Actually, I see this is done in figure 2. Consider eliminating Figure 1 and describing Figure 2 in its place.

We have carefully considered how best to address this, and opted for incorporating Reviewer 2's comments and suggestions to improve clarity in explaining collider bias to the uninitiated reader. We feel that the reviewer's earlier comment about collider bias being unfamiliar to many readers is important, and therefore having a visual that explicitly states how a risk factor and an outcome (Figure 1A) or underlying factors for a risk factor and an outcome (Figure 1B) can lead to sample selection importantly allows us to provide some intuition about the process in an accessible manner. More specifically to the present comment, Figure 2 provides specific examples of this relating to COVID research.

-References required for the second-last sentence of the second paragraph (The magnitude of this distortion...)

We have added the following references to substantiate this statement:

1. Elwert F, Winship C. Endogenous Selection Bias: The Problem of Conditioning on a Collider Variable. *Annu Rev Sociol.* 2014 Jul;40:31–53.
2. Nguyen, T., Dafoe, A., and Ogburn, E. L. (2019). The Magnitude and Direction of Collider Bias for Binary Variables. *Epidemiologic Methods* 8(1). <https://doi.org/10.1515/em-2017-0013>.

-Paragraphs 2,3,4 all start with “Suppose...”, a very colloquial term.

We have edited the text highlighted by the review and throughout the manuscript to avoid colloquial wording.

-There are already numerous well-known articles describing collider bias in the epidemiologic literature. I don't think Example 1 is necessary in this manuscript. It's not related to the topic being discussed. Would either revise to make it a COVID-specific example, or, remove.

As in our response to a previous point above, given that collider bias is likely to be unfamiliar to some readers we feel that it is important to demonstrate a 'lay' example. This also allows us to provide some intuition about the process in an accessible manner. We have therefore now removed explicit discussion of the lay example from the main text, and instead refer to Figure 1 to provide an accessible demonstration of collider bias. We believe this now strikes the right balance.

-Where is the discussion of figure 1B?

We have now included discussion of Figure 1B in the introduction. The new text reads: “Collider bias can be induced not only when the collider is directly caused by the variables, but also when they share common causes with the collider (Figure 1B). That is, collider bias can affect results even where variables of interest do not directly cause the collider.”

-In Example 2, why would the target population be all adults in a study of healthcare workers?

In Example 2, we wish to test the hypothesis that being a health worker is a risk factor for severe COVID-19 symptoms. To reliably estimate the association between being a healthcare worker and COVID-19 symptom severity, we require data on the total population of all adults (that is, all healthcare and non-healthcare workers). We could, of course, take a random sample of this target population, but the central point is that our target population is all adults, and health care work is the (occupational) exposure of interest.

-“To give context on how serious a problem collider bias can be, there is a continuing debate in the literature about the extent to which it is appropriate to adjust for covariates in observational associations”. This sentence, and the ones that follow, confuse me. Adjusting for a confounder will not induce bias, adjusting for a collider will. The quoted sentence is not correct- there is minimal debate on this topic- adjusting for a confounder closes a ‘backdoor path’ therefore removing bias, whereas conditioning on a collider induces bias. Please clarify what debate you feel exists in the literature.

We have now edited the text in this section to make this clearer. The new text now reads: “The extent to which it is appropriate to adjust for covariates in observational associations depends on where the covariates sit and their role in the data generating process (14–17). If we assume that a given covariate influences both the hypothesised risk factor and the outcome (a confounder), it is appropriate to condition on that covariate to remove bias induced by the confounding structure. However, if the covariate is a common consequence rather than a common cause (a collider), then conditioning on the covariate can induce, rather than reduce bias (18–20).”

-What is the “hidden causal structure”?

We have edited the text to make clearer what we mean. The new text now reads: “A priori knowledge of the underlying causal structure of variables and their role in the data generating process can be hard to infer, therefore it is appropriate to treat collider bias with a similar level of caution to confounding bias.”

-Box 3 and Figure 3 are unnecessary. Figure 3 should be eliminated. Recently, there has been extensive discussion about the role (or lack thereof) of p-values in epidemiologic analyses. I understand the point the authors are trying to convey about Factors influencing being tested, and think that if they choose to retain Box 3 the focus should be on discussing the specific variables that were strongly associated with being tested, not p-values (or inverse probability weighting, for that matter).

We understand the point the reviewer is making here, but we feel it is central to the manuscript to explain the extent to which tested samples are non-random. This links to what we feel is an important point that it can be very difficult to know what the sample selection model is. Performing a hypothesis-free scan feels like an appropriate way to achieve this. We fully agree that it is important to discuss

specific variables on the basis of the effect sizes without reliance on p-values when making epidemiological claims. To this end we have written a standalone document (Supplementary Note, available here: <https://mrcieu.github.io/ukbb-covid-collider/>) that walks through an example analysis for some of the variables in UK Biobank, in which we demonstrate the impact that selection based on these variables have on distorting relationships, and illustrate the coding steps required to implement some of the available methods to address the problem.

-Frailty is not synonymous with individuals who live in assisted living facilities/long term care. Frailty is a distinct construct, which may be present in some older adults who live in assisted living, but not all, and may be present in community dwelling older adults.

We have edited the text to make it clearer that frailty is not specific to older adults living in long-term care or assisted living facilities. The new text now reads: “Defined here as greater susceptibility to adverse COVID-19 outcomes, frailty is more likely to be present in certain groups of the population, such as older adults in long-term care or assisted living facilities, those with pre-existing medical conditions, obese groups, and smokers. These factors are likely to strongly predict hospitalisation. At the same time, COVID-19 infection and severity likely have an influence on hospitalisation (8–10,34), meaning investigating these factors within hospitalised patients may induce collider bias. Additionally, groups may be treated differently in terms of reporting on COVID-19 in different countries (49). For example, in the UK early reports of deaths “due to COVID-19” may have been conflated with deaths “while infected with COVID-19” (50). Individuals at high risk are more likely to be tested in general, but specific demographics at high risk such as those in long-term care or assisted living facilities have been less likely to be sampled by many studies (49). Frailty also predicts hospitalisation differentially across different groups, for instance, an older individual with very severe COVID-19 symptoms in an assisted living facility may not be taken to hospital where a younger individual would (51).”

-One of the most important aspects of this pandemic is racial/ethnic disparities in care, morbidity, and mortality. The token mention of ‘Ethnicity’ in the manuscript is inadequate to address this critically important issue.

Thank you for this suggestion. We have now restructured the section and provided a more substantial exploration of how ethnicity influences sample selection. The directly relevant text is copied below:

“Ethnic minorities are also more likely to be infected with COVID-19 (44). Adverse COVID-19 outcomes are considerably worse for individuals of some ethnic minorities (45). This could conceivably bias estimated associations within sampling frames based within hospitalised patients, as in many countries, ethnic minority groups are over-represented as ethnic inequalities in health are pervasive and well-documented. Furthermore, ethnic minority groups are more likely to be key workers, who are more likely to be exposed to COVID-19 (46). Cultural environment (including systemic racism) and language barriers may negatively affect entry into studies, both based on testing and voluntary participation (47). Ethnic minority groups may be more difficult to recruit into studies, even within a given area (48), and may affect the representativeness of the sample. Ethnic minorities were less likely to report being tested in our analysis of the UK Biobank data, where one of the strongest factors associated with being tested was the first genetic principal component, which is a marker for ancestry (Box 3). Thus, this could present as above, with ethnic minorities’ presentation to medical care being conditional on more severe symptoms.”

-“The primary task in any analysis is to evaluate the extent to which sample selection is likely to have actually occurred”. I don’t believe this is correct. Isn’t the primary task of an analysis to evaluate the exposure-outcome relationship of interest?

We wished to convey that this was the primary task for methods to either overcome or evaluate how sensitive associations are to collider bias. We realise that our text was confusing and have now edited out this statement:

“In this section we describe methods to either address collider bias or evaluate the sensitivity of results to collider bias. As with confounding bias, it is generally not possible to prove that any of the methods has overcome collider bias. Therefore, sensitivity analyses are crucial in examining the robustness of conclusions to plausible selection mechanisms (18,19).”

-There are no references included in the first three paragraphs on “methods for overcoming bias”. This must be corrected.

We have added the following references to substantiate this statement:

- 1. Nguyen TQ, Dafoe A, Ogburn EL. The magnitude and direction of collider bias for binary variables [Internet]. arXiv [stat.ME]. 2016*
- 2. Ding P, Miratrix LW. To Adjust or Not to Adjust? Sensitivity Analysis of M-Bias and Butterfly-Bias. Journal of Causal Inference. 2015 Mar 1;3(1):41–57.*
- 3. Munafò MR, Tilling K, Taylor AE, Evans DM, Davey Smith G. Collider scope: when selection bias can substantially influence observed associations. Int J Epidemiol. 2018 Feb 1;47(1):226–35.*
- 4. Bareinboim E, Tian J, Pearl J. Recovering from Selection Bias in Causal and*

Statistical Inference. In: Proceedings of the Twenty-Eighth AAAI Conference on Artificial Intelligence. Québec City, Québec, Canada: AAAI Press; 2014. p. 2410–6. (AAAI'14).

-What are “subtle departures in the characteristics of the study sample”? How is this evaluated? What defines subtle?

We have edited the text to make clear that we are referring to any differences between the sample and the general population. The new text now reads: “If there are departures in the characteristics of the study sample from the general population then this provides evidence of selective sampling, even where a random sampling approach was used”.

-Is it ever possible to prove the absence of selection through validation?

We don't think it is practically possible to do this, as there could be unmeasured factors influencing selection, or interactions between factors that are not detected when comparing only marginal distributions, meaning that one would require all data that relate to selection. Furthermore, all data would require to be measured without bias and only random measurement error, which in practice is implausible. We have added a section to clarify this, with the following text:

“While it is encouraging if the sample estimates match their population-level equivalents, it is important to recognise that this does not definitively prove the absence of collider bias (54). This is because factors influencing selection could be unmeasured in the study, or factors interact to influence selection and go undetected when comparing marginal distributions.”.

-Are sensitivity analyses a method for overcoming collider bias? The title of the final section of the manuscript should be revised.

We have now renamed the title of this section to “Methods for detecting and minimising the effects of collider bias”.

-Isn't it possible to use a form of inverse weighting (weighting by the inverse of selection probabilities) in non-nested samples? See Tim Lash's work on bias analysis for a thorough description.

Yes this is certainly possible (e.g. Tudball et al 2020) and is something we discuss in the section that is now titled 'Non-nested samples', and provide an example of how to perform the analysis in the Supplementary Note

-I understand the point the authors are trying to convey when describing a situation “where the entire dataset comprises only the selected samples used for hypothesis testing (stand-alone)”. However, I think the description provided may be confusing to the reader who is not as familiar with bias analysis techniques. I think many people would think “don't we always use all our data available for analyses?”. Also, as described in the text, external information is required for the bounding approach so it's not completely stand alone. Perhaps focusing on Nested vs. Non-nested samples would be more clear.

Thank you for this suggestion; we have now rephrased “stand-alone” samples as “non-nested”. We have also elaborated on the text in this section to make the description of non-nested samples clearer to readers. The new text reads:

“The applicability of different methods crucially depends on the data that are available on non-participants. These methods can broadly be split into two categories based on the available data: nested and non-nested samples. A nested sample refers to the situation when key variables are only measured within a subset of an otherwise representative sample, thus forcing analysis to be restricted to this sub-sample. An example close to this definition is the sub-sample of individuals who have received a COVID-19 test nested within the UK Biobank cohort (though, it is clear that the UK Biobank cohort is itself non-randomly sampled (15)). For nested samples researchers can take advantage of the data available in the representative super-sample. A non-nested sample refers to the situation when data are only available in an unrepresentative sample. An example of this is samples of hospitalized individuals, in which no data are available on non-hospitalized individuals. It is typically more challenging to address collider bias in non-nested samples.”

-There is a Figure 4 described in the Text but I don't see it in the manuscript file.

We had incorrectly labelled two figures as “Figure 3” and have now rectified this.

REVIEWERS' COMMENTS

Reviewer #1 (Remarks to the Author):

The authors have done a good job to address my comments.

Please check reference numbers, e.g. page 10 lists reference (55) linking to Zhao et al., but I think this should be reference (104).

Though not directly relevant for the submission, I continue to find Ascertain to be somewhat vague. For instance, in the 'Estimation' part, the figure has no labels. Furthermore, expressions like 'differential effect on the probability', which are used left of the plot, are vague as it is unclear on what scale the effect is measured. Although this is explained in the theory part, why not refer to 'differences in probability'?